# Research on intelligent routing with VAE-GAN in the internet of body

**Song Qian[1], Tianping Zhang[2]\*, Siping Hu[3]**

**1** School of Information Engineering, Xinjiang Institute of Technology, Xinjiang, China, **2** School of Mathematics and Computer Science, Hanjiang Normal University, Shiyan, China, **3** Institute of Engineering and Technology, Hubei University of Science and Technology, Xianning, China

\* zhangtianping0818@163.com

**Data Availability Statement:** Data cannot be shared publicly because of muman body data parameters. Data are available from the Local hospital Institutional Data Access / Ethics Committee (contact via 1273091366@qq.com) for

## Abstract

The "Internet of Body" is an emerging technology that is centered on the human body and connected to the Internet. It can monitor a variety of human data (such as heart rate, blood oxygen content, etc.) and communicate with digital pills, wearable devices, etc. It has been widely used in the field of medical health. However, when other devices access the Internet of Body on a large scale, there will be load imbalance caused by the difficulty in selecting the optimal route, which will affect the overall throughput and may even fail to transmit and endanger life. The traditional artificial intelligence routing algorithm cannot deal with the low model prediction accuracy and poor generalization ability caused by large noise and small data volume. This paper proposes an artificial intelligence routing algorithm, combines the variational autoencoder (VAE) and the generative adversarial network model (GAN) to construct a VAE-GAN model to generate multiple sets of data to achieve data enhancement on the Internet of Body. The optimization goals are to maximize the throughput of the Internet of Body and minimize the transmission cost. The entire routing problem is expressed as a Markov decision and the optimal transmission path is solved by learning previous historical experience to generate the real-time optimal route. Experiments have shown that this scheme can achieve the optimal route according to the transmission capacity of the real-time path and only requires fewer computing resources. It achieves load balancing of the entire network and avoids network congestion. The average throughput is much higher than that of traditional routing, and the advantage is more obvious under high load.

## 1.Introduction

The Internet of Bodies (IoB), also known as the Internet of Things based on the human body, is an emerging technology that connects the human body directly to the Internet and collects and processes human data through built-in or wearable devices. This technology is mainly used in the field of medical health, such as monitoring and analyzing human health through implantable pacemakers, ingestible digital pills, wearable prostheses, and other devices. The collected data includes steps, heart rate, blood oxygen content, etc. [1], as shown in Fig 1. In recent years, with the rapid development of a variety of built-in or wearable devices, the amount of information communicated with the IoB has increased rapidly, and the number of

researchers who meet the criteria for access to confidential data.

**Funding:** The author(s) received no specific funding for this work.

**Competing interests:** The authors have declared that no competing interests exist.

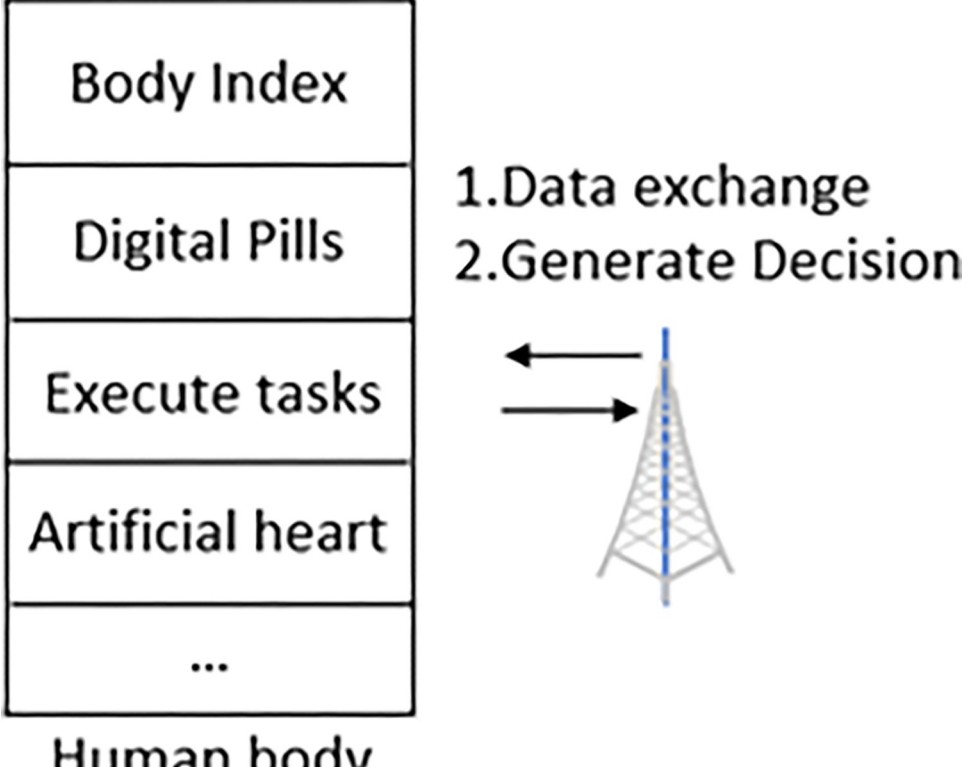

**Fig 1. Body-connected routing scenario.**

deployment types has increased. The load imbalance caused by path selection under high concurrency has become the primary factor restricting the efficient operation of the IoB. Therefore, it is necessary to study intelligent routing algorithms that are more suitable for the IoB under large-scale access.

The Internet of Things mainly consists of the perception layer, network layer, and application layer, among which routing plays a key role in the entire communication process. Most of the current routing modes are to implement network interoperability between servers. Routing is composed of multiple selected paths. Under normal access scale, only the shortest path routing solution is required. However, when there is large-scale access When used in applications in the Internet of Things (such as an increase in the amount of data sensed in a short period of time), performance degradation will occur. Although shortest paths routing is a routing scheme that selects the smallest number of paths and is suitable for situations where the overall transmission performance is good, it is not the optimal routing scheme in all cases. The current artificial intelligence routing algorithm has the following problems:

Poor model adaptability: Since the Internet of Things is a network deployed in the human body, it is affected by many aspects and cannot carry out regular model training like the Internet of Things, and the model adaptability is poor.

Sparse sample data: The training method based on deep learning requires a certain amount of data as support. However, the Internet of Things is affected by many aspects, and the training data is small, so it is difficult to obtain valuable samples.

Low intelligence: Existing routing algorithms have the disadvantage of low intelligence in the Internet of Things. With the emergence of technologies such as cloud computing and high concurrency, there is a need for more intelligent routing algorithms to adapt to them.

In order to solve the above problems, this article analyzes the key technologies of IoT routing, takes maximizing the overall IoT transmission throughput and minimizing transmission cost as optimization goals, constructs a VAE-GAN model to achieve data enhancement, and then proposes an intelligent routing based on transfer Learning, design real-time optimal intelligent routing scheme.

## 2. Related work

The Dijkstra algorithm is a commonly used routing algorithm in the network. It stores all vertices and calculates edge values. When it is used again, it can be directly queried. It needs to maintain the shortest path table. When the access scale is large, the search efficiency from the path table is low and load balancing cannot be achieved well.

Equal-Cost Multipath Path (ECMP) equal-cost routing algorithm stores all des t IPs and paths with the same cost value. When used again, it can be queried directly. The advantage is that it is easy to deploy, and the load is more balanced than other algorithms. The disadvantage is that all Routes are distributed in equal parts, which cannot satisfy services with high immediacy requirements.

K -shortest Path (KSP) is an extension of the Dijkstra algorithm. It excludes intersecting paths and paths that may form loops. It also needs to maintain the shortest path table. It has high requirements on hardware, is difficult to deploy, and is not easy to achieve load balancing.

Dynamic Scheduling Load Balancing (DSLB) dynamic scheduling load balancing algorithm. The principle of this algorithm is to greedily find the maximum available bandwidth. It is similar to polling routing. When ensuring the maximum bandwidth, polling uses available paths without affecting delay, Judgment of other performance indicators such as available bandwidth, so there are limitations. In addition, other scholars have also conducted research on cloud computing data center network routing. There are many routing implementation methods for cloud computing data center networks. This article mainly discusses routing solutions that can ensure system load balancing under large-scale access. When accessing the Internet of Things on a large scale, in order to prevent load imbalance, all key factors need to be considered comprehensively and comprehensively, and routing algorithms based on artificial intelligence have great advantages in this regard.

Since the Internet of Body and the Internet of Things have many similarities (such as dynamic network topology), there are relatively few literatures on the Internet of Body. Therefore, in many aspects, we can refer to the routing of the Internet of Things, the Internet of Water, etc. For example, in 2024, Zhu et al. designed an underwater routing protocol to deal with the drawbacks of high propagation delay and dynamic network topology in the Internet of Water [2]. In 2023, Malik et al. proposed a routing algorithm based on reinforcement learning to reduce end-to-end routing delay in the Internet of Things and improve throughput. Experiments have shown that it has advantages such as reducing packet conflicts [3]. In 2020, Kaur et al. proposed an artificial intelligence-based routing algorithm for the heterogeneity of nodes in the Internet of Things. Experiments have shown that it improves stability [4].

In 2024, Arafat et al. designed an adaptive Q-learning adaptive routing algorithm to address the frequent changes in network topology required for monitoring personal physiological parameters [5]. In wireless body area networks, in order to achieve reliable routing, artificial intelligence multi-source aggregation routing is proposed, which is beneficial for avoiding transmission interruptions caused by congestion [6]. In 2021, V Degli Esposti et al. summarized the transmission and channel modeling of millimeter wave and terahertz waves [7]. In 2024, Liu et al. proposed a routing algorithm with multiple DQNs to generate access order and

the number of health data packets sent for each node in order to achieve early disease monitoring and balance waiting time. The experiment proved that it has good performance in asymmetric systems [8]. Li et al. proposed a real-time differential multimodal transformer that helps collect real-time human life parameters [9]. Vakil et al. proposed sample collection steps in different environments to address the problem of limited prediction samples, which can help train better models [10].

Transfer learning is a machine learning method that takes the model developed for *task*1 as an initial point and reuses it in the process of developing the model for *task*2. It means transferring knowledge from one domain (source domain) to another domain (target domain) to accelerate the learning process of the new domain. This ability to draw analogies is not only in line with human learning laws, but also one of the important goals pursued by artificial intelligence. In deep learning, transfer learning achieves fast adaptation and efficient learning by utilizing pre trained models and combining them with a small amount of new domain data.

Routing is a sequential decision problem, in which each decision is a function of the current state of the network. Therefore, routing can be modeled as a Markov decision process, which is naturally applicable to the field of artificial intelligence. Transfer learning is generally speaking about designing an algorithm to enable it to have the ability to draw inferences from one instance, that is, to learn new knowledge by applying existing knowledge. Its core is to find the similarities between existing knowledge and new knowledge, and to achieve the purpose of transfer learning through the transfer of this similarity. Since many things have commonalities, how to reasonably find the similarities between them and then use this bridge to help learn new knowledge is the core problem of transfer learning. Its working principle is shown in Fig 2.

Transfer learning usually focuses on a source domain $Data_s = \{a_i^s, b_i^s\}$ and a target domain $Data_t = \{a_i^t, b_i^t\}$, which $a_i^d, b_i^d$ respectively represent data samples and corresponding category labels. The definition of transfer learning is: given the source domain $Data_d$ and learning task $T_s$, the target domain $Data_t$ and the learning task $T_t$, the purpose of transfer learning is the acquisition $Data_s$ of knowledge in the source domain $f_t(\cdot)$ and learning tasks to help improve the learning of prediction functions $T_s$ in the target domain, where $Data_s \neq Data_t$ and $T_s \neq T_t$.

Commonly used generative models in the field of deep learning include VAE and GAN [11]. Although both can perform data regeneration, each has certain flaws in its application. The VAE model deep model and static reasoning can obtain the distribution of the original data through resampling, but the diversity of the generated data is low; the GAN model

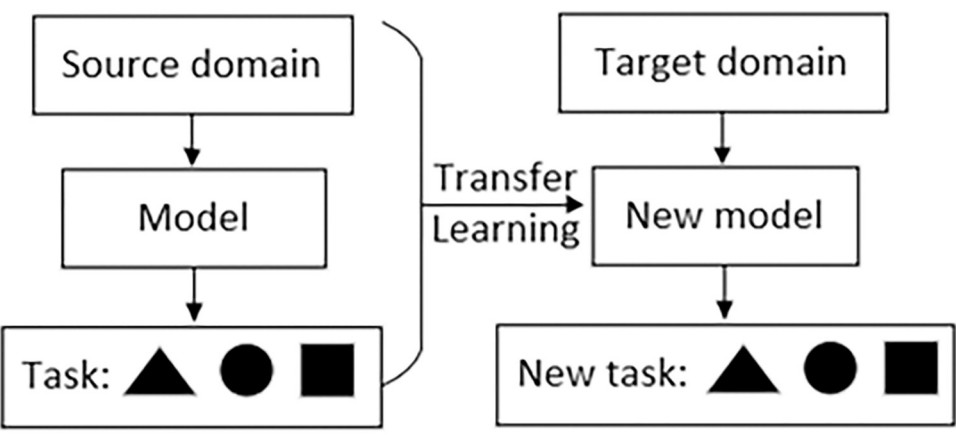

**Fig 2. Principle of transfer learning.**

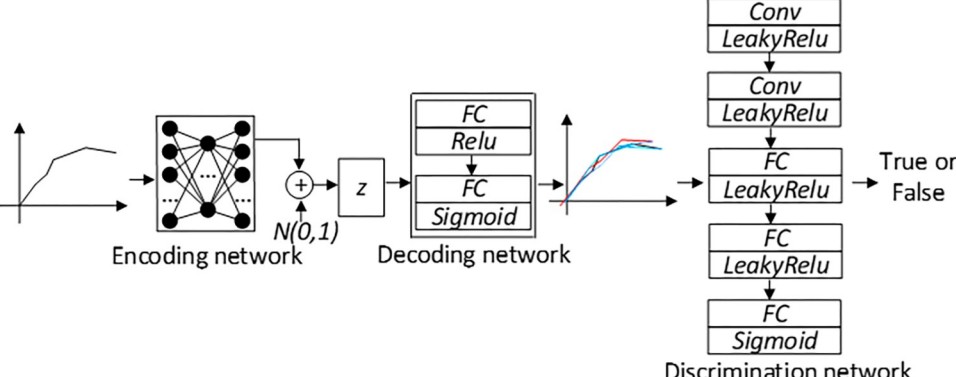

**Fig 3. VGR structure.**

generates more diverse data, but the training process is extremely difficult, and it is difficult for the generator and the discriminator to simultaneously convergence. Therefore, this article combines the data generation of the VAE model and the adversarial learning mechanism of the GAN model to propose the VGR (VAE-GAN-route) model, taking into account the advantages of both and giving full play to their respective strengths. The VAE network is used as a generator based on the GAN network, which optimizes the GAN generator's shortcomings of single extraction features, low cohesion, and easy gradient disappearance. At the same time, the GAN network can make up for the lack of sample data due to VAE's lack of adversarial learning, improving the performance of the two alone. shortcomings in the work, resulting in higher accuracy and robustness of reconstructed data.

The proposed VGR model is divided into three parts, namely the encoding network, the decoding network (generating network) and the judging network, which correspond to the Encoder and Decoder of the VAE model and the Discriminator of the GAN model. It can be understood as adding the encoding process of real data on the basis of the GAN network, using the implicit z with prior information to replace the random vector originally input to the generator, which greatly improves the expression ability of the model. The VGR model structure is shown in Fig 3.

The encoding network consists of three fully connected layers. The first fully connected layer down samples the input high-dimensional data, and the activation function uses Relu; the remaining two fully connected layers map the down sampled samples into two low-dimensional parameter features, namely the mean and variance of the normal distribution. The probability distribution of the original sample data $x$ is:

$$x \sim P_\theta(x) \tag{1}$$

In the above formula, $\theta$ is the model parameter. The loss function of the encoding network is KL (Kull back Libeler Divergence) divergence, used to measure the difference between the implicit vector distribution and the standard normal distribution. The KL loss function is:

$$KL(N(\mu, \sigma^2) \parallel N(0, 1)) = \frac{1}{2} \left[ \log_2(\sigma^2) + \sigma^2 - \mu^2 - 1 \right] \tag{2}$$

In the above formula, $\mu$ and $\sigma^2$ are respectively the mean and variance of the encoding network output. Indicates the calculation of the KL divergence of two approximate distributions. It is used here to express the KL divergence between the data distribution sought in this article and the standard normal distribution $N(0,1)$. If the two distributions are the same, the KL

divergence is 0. On the contrary, the greater the distribution difference, the greater the KL divergence value. Variational Encoders (VAE) are based on Bayesian inference and their goal is to model latent and sample new data in the model, while Generative Adversarial Networks (GAN) are based on game theory and their goal is to find Nash Balanced discriminator and generator networks.

KL divergence is a concept in statistics, a procedure used to measure the similarity of two probability distributions. When the two probability distributions are closer, their KL divergence value becomes smaller. For discrete probability distributions, the definition is as follows:

$$D_{KL}(P||Q) = \sum_i P(i) \log \frac{P(i)}{Q(i)} \tag{3}$$

$P$ and $Q$ is a probability function the variable $i$. And the continuous probability distribution is:

$$D_{KL}(P||Q) = \int_{-\infty}^{+\infty} \log \frac{p(x)}{q(x)} dx \tag{4}$$

Therefore, if we want to pass a random Gaussian noise $z$ through a generative network G to obtain a $Pdata(x)$ generative distribution that is similar to the real data distribution $PG(x, \theta)$, where the parameters $\theta$ are determined by the parameters of the network, we hope to find a generative distribution $\theta$ that is as close $Pdata(x)$ as possible to $PG(x, \theta)$.

The decoding network (generating network) consists of two layers of fully connected networks. The parameter settings of these two layers of network are opposite to the parameter settings of the encoding network. The input low-dimensional hidden variables are characterized by amplification, and finally generated data of the same size as the input data is generated., the Relu activation function is used between the two network layers, and the final output layer uses the Sigmoid activation function. The optimization goal of the generation network must not only ensure that the feature loss between the generated samples and the real samples is minimized but also ensure that the generated data can deceive the discriminant network. The loss function is:

$$L_G = (x_i - \hat{x}_i)^2 + E_{z \sim p(z)} [\log_2 (1 - D(G(z)))] \tag{5}$$

In the formula, the first term on the right side represents the loss reconstruction of the encoder, $x_i$ and $\hat{x}_i$ are the real samples and predicted values. All hidden variables generated for parameter reconstruction obey $p(z)$ distribution. $G(z)$ is used to generate samples; $D(\cdot)$ is used to judge the authenticity of the generated data by the discriminating network.

The discriminant network consists of 2 convolutional layers and 3 fully connected layers. The activation function between each network layer uses LeakyRelu. The output layer outputs the authenticity label through the Sigmoid activation function. The loss function of the discriminant network uses a more stable least square. The multiplicative loss function is:

$$L_D = \frac{1}{2} E_{x \sim p_{data}(x)} [(G(x))^2] + \frac{1}{2} E_{x \sim p(z)} [(D(z) - 1)^2] \tag{6}$$

In the formula, $x \sim p_{data}(x)$ means that the real sample obeys $p_{data}(x)$ distribution.

The VAE-GAN model takes advantage of the coding advantages of VAE to input hidden vectors containing prior information into the GAN model instead of random vectors for data generation, thereby alleviating the problem of non-convergence in model training. The GAN network improves the quality of generated data through adversarial learning and completes the improvement of the VAE model. The two complement each other.

## 3 Intelligent routing algorithms

### 3.1 The multi-objective optimization model

The current intelligent routing is reflected in the selection of the best path for each routing decision. Intelligent routing takes into account the load balancing of the entire network, and it also makes the best use of the available transmission paths (bandwidth and number). Furthermore, due to the characteristics of the Internet of Body, the shortest path is not the optimal transmission path, and the real-time transmission path is changeable. Therefore, this paper studies the transmission performance and effective bandwidth of the path and designs a multi-objective optimization intelligent routing algorithm based on transfer learning. In the Internet of Body, the entire network is modeled as $G = (V, N)$, where $V$ is a non-empty vertex set, $|V|$ is the number of vertices. $N = \{v_i|v_j, i \neq j, v_i, v_j \in V\}$ is an edge set and $v_i|v_j$ is the communication link between nodes $v_i$ and $v_j$ [12].

Furthermore, $B$ is the residual bandwidth set. The residual bandwidth of an edge $v_i|v_j$ is expressed as $b_{i,j} \in B(i \neq j, i \geq 1, j \geq 1)$, and $\forall v \in V$, $b_{i,i} = \infty$. The following definition can be obtained:

Definition 1: A route $a_{s,t}$ is defined as the edges from a source node $v_s$ to another target node $v_t$:

$$a = \{v_s|v_1, v_1|v_2, \cdots, v_{m-1}|v_m, v_m|v_t\} \tag{7}$$

Among them, $v_s \neq v_1$, $v_m \neq v_t$, and for $\forall v_i, v_j$, if $i \neq j$, then $v_i \neq v_j$, based on the definition of routing, we further define that $k$ disjoint paths from the source node to the target node are:

$$A_{s,t}^k = \{a_{s,t}^i\}_{i=1}^k \tag{8}$$

In the above expression, $\forall i \neq j, A_{s,t}^i \cap A_{s,t}^j = \emptyset (i \geq 1, j \geq 1)$.

Definition 2: The bottleneck bandwidth of the route $a_{s,t}$ from the source node $v_s$ to the target node $v_t$ is:

$$BB(a_{s,t}) = min(b_{s,1}, b_{1,2}, \cdots, b_{m-1,m}, b_{m,t}) \tag{9}$$

There are multiple paths $A_{s,t}^k$ from the source node $v_s$ to the destination node $v_t$, and the maximum bottleneck bandwidth (MBB) of the path $a_{s,t}$ is $A_{s,t}^k$. The maximum available bottleneck bandwidth on all paths, that is:

$$MBB(A_{s,t}^k) = max(BB(a), BB(a_{s,t}^2), \cdots, BB(a_{s,t}^k)) \tag{10}$$

The minimum hop count (MHC) from the source node $v_s$ to the destination node $v_t$ is the minimum hop count of all nodes:

$$MHC(A_{s,t}^k) = min(|a_{s,t}^1|, |a_{s,t}^2|, \cdots, |a_{s,t}^k|) \tag{11}$$

Where, $|a_{s,t}^1|$ represents the number of elements in the set $a_{s,t}^1$.

The path transmission delay $a_{s,t}$ from the source node $v_s$ to the destination node $v_t$ is defined as follows (sum of link delays):

$$PD(a_{s,t}) = d_{s,1} + \sum_{i=1}^{m-1} d_{i,i+1} + d_{m,t} \tag{12}$$

Then, the minimum propagation delay $a_{s,t}$ of the path is the minimum propagation delay of all

paths $P_{s,t}^k$, that is:

$$MPD(a_{s,t}^k) = min(PD(a_{s,t}^1), PD(a_{s,t}^2), \cdots, PD(a_{s,t}^k)) \tag{13}$$

The $k$ disjoint paths from node $v_s$ to target node $v_t$ can be expressed as a multi-objective optimization model:

$$Max(MBB(A_{s,t}^k)) \tag{14A}$$

$$Min(MHC(A_{s,t}^k)) \tag{14B}$$

constraint:

$$\forall a_{s,t}^i, a_{s,t}^j \in A_{s,t}^k, a_{s,t}^i \cap a_{s,t}^j = \emptyset \tag{15A}$$

$$k \geq 1 \tag{15B}$$

## 3.2 Algorithm process

The implementation process of the multi-objective optimization model in this paper is improved on the basis of literature [13–15]. The entire algorithm implementation process is as pseudo code Algorithm 1. The former part is the data enhancement part, and the next part outputs the final result.

The time complexity of the algorithm is not high [16], and it can achieve unified processing of multiple goals as a whole. Next, the algorithm is simulated to verify its effectiveness and evaluate its performance.

```
Algorithm 1 Intelligent routing algorithm under high load of Internet
of body
Input Current routing data, current network status, etc.
Output Next optimal route
01 A = collecting all routing data, and get the training data and test-
ing data
02 for a! = null and a∈A do
03    create VAE-GAN model
04    put all routing data a to VAE-GAN model and train data
05    if is_max_train_number then
06    save the model WS, and get all the created data
07    end if
08    liding window segmentation data
09 end for
10 B = get all the test data
11 for b! = null and b∈B do
12    put b to model WS
13    if is_max_train_number then
14    regression prediction
15    end if
16 end for
17 Return the prediction result
```

Among them, lines 01–09 are the data addition part. Lines 10–17 are the processing of input test data. *is_max_train_number* is the maximum training frequency identifier. The designed algorithm focuses on low computation and space complexity and is suitable for application in the Internet of body. The entire algorithm consists of 2 loops, with the number of loops being $tn1$ (lines 02–09) and $tn2$ (lines 11–16) respectively. The time complexity of the entire algorithm is $O(tn1)+O(tn2)$. If the number of loops is $n$, then the time complexity of the algorithm is $O(n)$.

## 4 Experimental simulation and result analysis

By building an Internet of Body and simulating access conditions under high load. Based on the simulation parameters in reference [17], the following simulation platform was built:

Hardware environment: Intel(R) Core (TM) i7 12400F CPU @2.50 GHz × 6, 64 G memory.

Python version number: Python 3.12.

Operating system: Ubuntu 24.04.

Network routing analysis and debugging tool: Wireshark v 4.2.4, used to analyze network routing, etc.

Load stress software: webbench 5.0 [18], this software can simulate tens of thousands of concurrent requests per second, support the setting of control time, cache and server response time, etc. Its multi-concurrency principle is that there are multiple subprocesses [19].

Comparing several commonly used routing algorithms in the experiment, the details are as follows:

ECMP: Equal cost multi-path routing, and its advantages are mainly reflected in the ability to improve network redundancy and reliability, while also increasing network resource utilization, detailed in reference [20].

Dijkstra: The shortest path algorithm selects the route with the minimum hop count, as detailed in reference [21].

WOAD3QN-RP, a routing algorithm based on swarm intelligence and deep reinforcement learning, is detailed in reference [22].

RL, a deep reinforcement learning routing algorithm, detailed in reference [23].

Proposed algorithm: the algorithm proposed in this article.

The comprehensive performance of various algorithms is analyzed by simulation comparison of the following indicators.

### 4.1. Normalized throughput

Throughput is used to measure the transmission capacity of a network [24]. It is calculated by the amount of data transmitted per unit time. Time is the time when data transmission starts and ends, and data is the total number of bytes of data transmitted during this period of time.

$$throughput = \frac{data_{success}}{time_{end} - time_{start}} \tag{16}$$

Throughput reflects the transmission capacity of the entire network. The transmission results are shown in Fig 4.

When the number of visits is average, the throughput of several algorithms is similar. As the number of visits increases, the algorithm in this article has the highest throughput, and it has been in stable transmission without large jitter. Since the Internet of Body is in dynamic changes in the link, it is easy to form a dynamic topology, so ordinary routing will have large throughput jitter, as shown in the figure, but the intelligent routing algorithm proposed in this article can easily cope with it. Methods such as Dijkstra and ECMP are not suitable for applications when the load is too high. They cannot cope with high access and high load well. The performance of ECMP is also average and not as good as the algorithm in this article. The

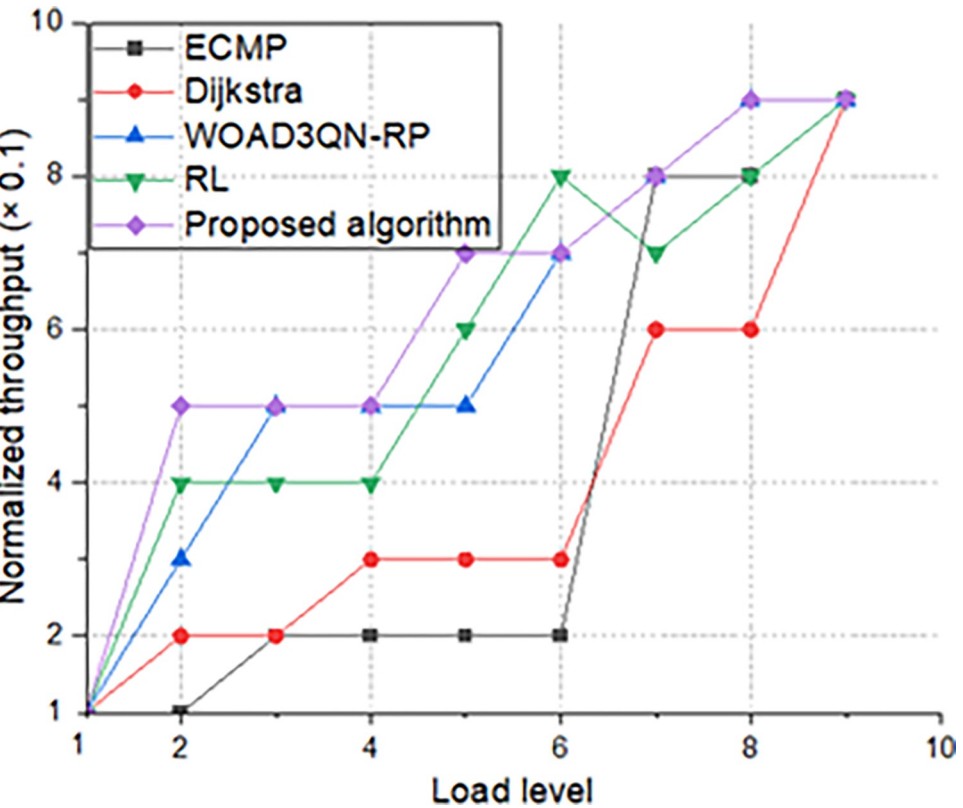

**Fig 4. Comparison of normalized throughput and load.**

algorithm in this article takes maximizing throughput as one of the optimization goals, and can the optimal routing plan is adjusted in real time. When load = 7, congestion is avoided without causing a significant decrease in throughput, so the overall transmission capacity is optimized by other algorithms. The normalized throughput tends to stabilize, indicating that the model can apply the influence of high dynamics.

## 4.2. Load and average bandwidth utilization

The software used for load concurrent generation is webbench 5.0. Average bandwidth utilization is calculated as:

$$Bu = \frac{\sum \frac{B_s}{B_r}}{x} \tag{17}$$

Among them, for a certain data flow in the Internet of body, $B_s$ is the bandwidth before sending, $B_r$ is the real-time bandwidth, and $x$ is the total number of statistical flows [25].

The relationship between load and average bandwidth utilization can directly reflect the performance during large-scale access, such as whether it is congested and whether it can meet high-load transmission, etc. The results are shown in Fig 5.

The comparison of average bandwidth utilization and load is shown in Fig 5. It can be concluded that when the access is of general scale, the average bandwidth utilization of all algorithms is about 80%. However, as the access scale increases, the average bandwidth utilization of the Dijkstra algorithm decreases first, and the rate of decrease is also faster than other algorithms and finally stops at 0.4. The Dijkstra algorithm has similar results. In addition, the two

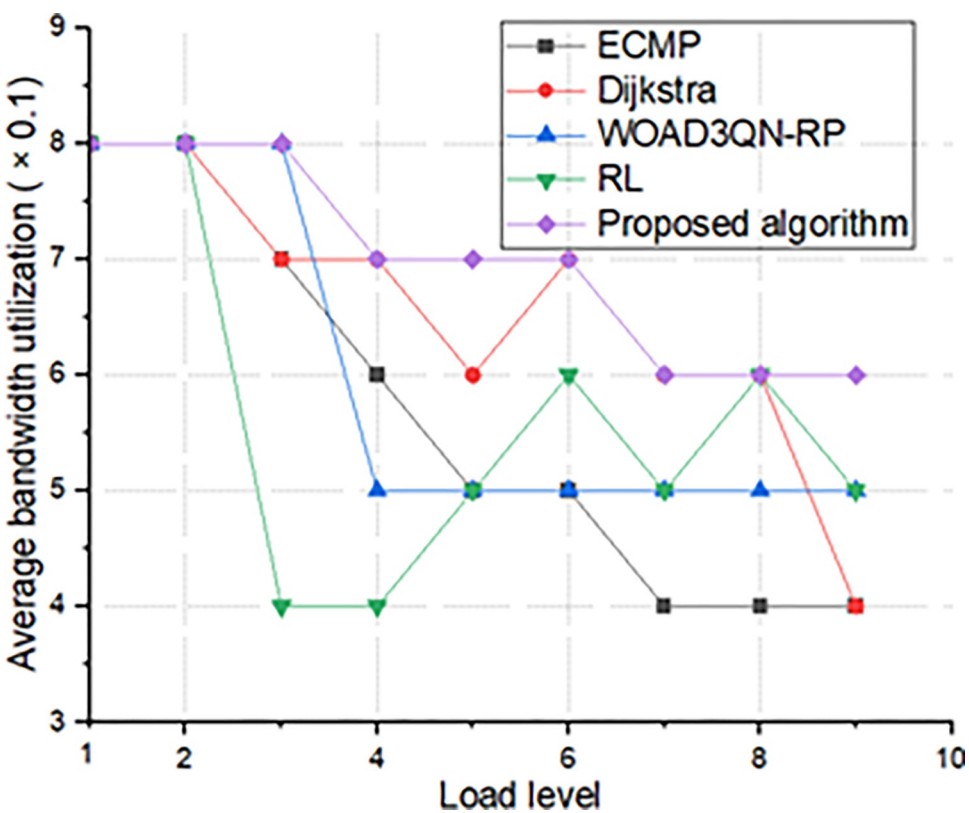

**Fig 5. Average bandwidth utilization and load comparison chart.**

algorithms cannot cope with high loads, so they are unstable and fluctuate greatly. ECMP has the ability to divide the bandwidth equally, so the average bandwidth utilization is better than the Dijkstra algorithm. The intelligent routing algorithm selects the optimal route according to the real-time network status based on the self-learning ability, which can better cope with high load and large-scale access, has a higher average bandwidth utilization, and has the best transmission performance.

The experiment shows that as the amount of data transmitted increases, the transmission time of several algorithms also increases. The result is shown in Fig 6. The transmission time of the method proposed in this paper is shorter than that of the other two algorithms, and it can cope with file transfer under high load.

From the above experimental data, we know that the possible shortcoming is that sporadic underestimation may occur during the training process, which affects the decision-making of reinforcement learning [26]. The solution is to apply multiple different neural networks to minimize the impact of underestimation on the core of the algorithm. Or the Bootstrapping method uses limited sample data to generate new, representative parent data by redesigning the sampling rules. Since the sampling rules are redesigned, it can also be recognized as a new sample, thereby expanding the training data and realizing the number of information training required can be completely ignored because the underestimation problem has a small impact on this algorithm. Furthermore, transfer learning solutions require a lot of trial and error and iteration, which this article studies may lead to poor solution results or over-fitting problems. As the number of transmitted files continues to increase, the proposed algorithm maintains stable transmission time.

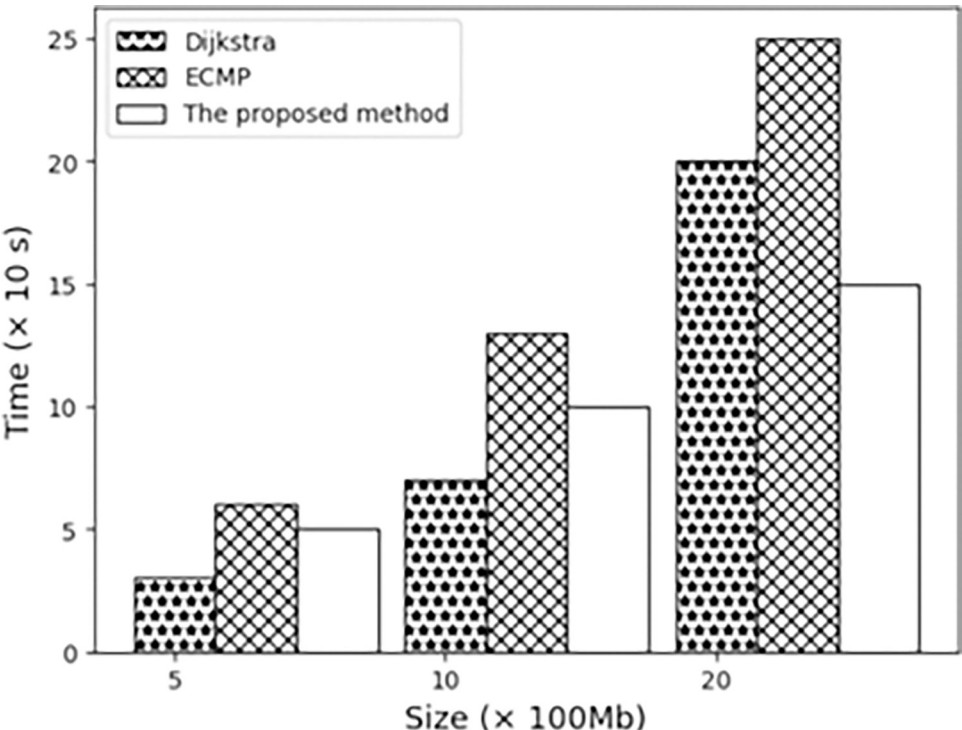

**Fig 6. Transmission time under different transmission numbers.**

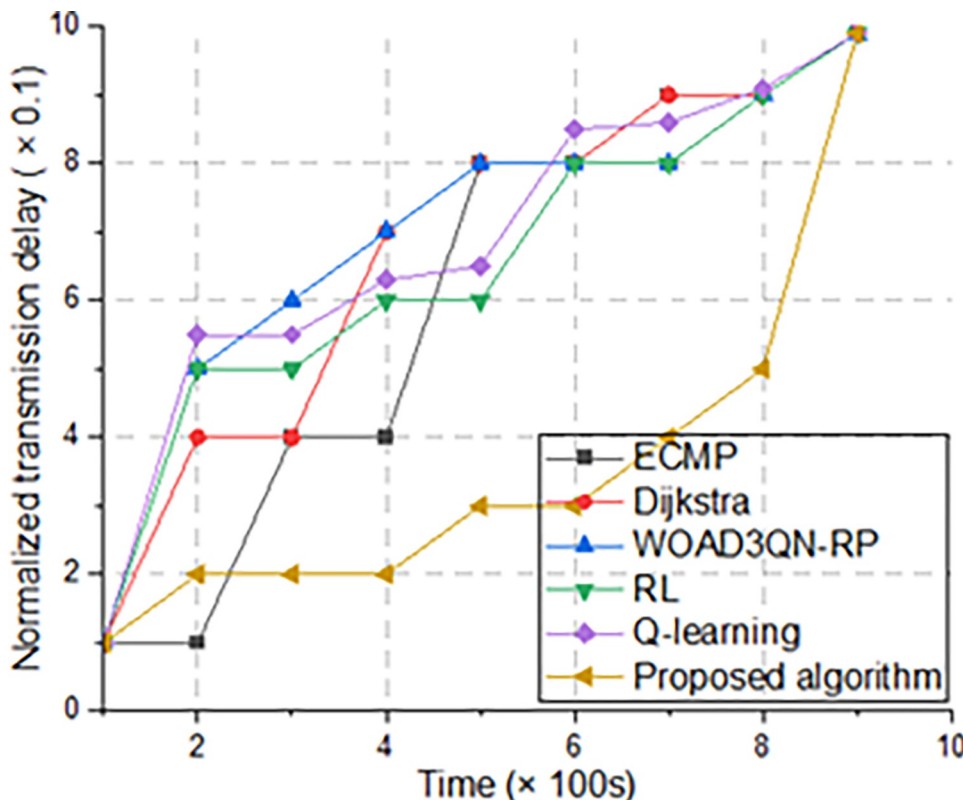

**Fig 7. Normalized transmission delay with time.** Q-learning is similar to RL, but its disadvantage is that it is not good at handling complex data. As shown in the figure, Q-learning has a higher transmission delay than RL.

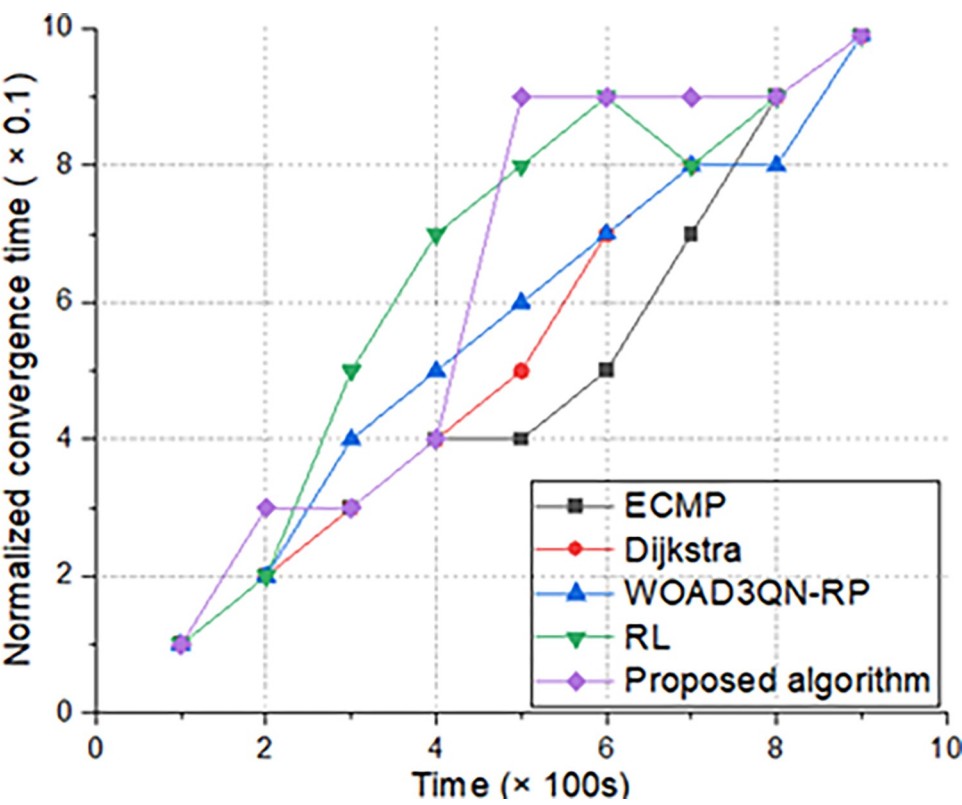

**Fig 8. Normalized convergence time.**

## 4.3. Normalized transmission delay

Normalized transmission delay refers to the transmission time of several algorithms for transmitting video files of the same size [27]. This experiment can demonstrate the ability to comprehensively transmit data. The algorithm proposed in this article can cope with congestion, dynamic topology changes, etc., so the transmission delay is the lowest. ECMP does not have the ability to handle dynamic topology, so transmission interruptions may occur and cause network congestion, resulting in the longest transmission delay. Both reinforcement learning algorithms also do not have the ability to handle dynamic topology, so the delay is lower than the algorithm proposed, as shown in Fig 7.

## 4.4. Normalized convergence time

Normalized convergence time refers to the convergence time of artificial intelligence algorithms trained on the same data [28]. The algorithm proposed applies transfer learning to solve transmission problems without overly relying on sample size, making it very suitable for applications similar to the Internet of body. However, the other two artificial intelligence algorithms have high time and space complexity, slow convergence, and are not suitable for solving routing problems in the "Internet of body ", as shown in Fig 8.

## 5 Conclusion

The Internet of Things has broad application prospects in the field of healthcare, work and life. This paper studies the load imbalance problem caused by the difficulty of routing

implementation when the Internet of Things network is accessed on a large scale. It proposes an intelligent routing algorithm for cloud computing data center networks based on reinforcement learning and realizes the solution of selecting the optimal route according to the real-time path transmission capacity. Through simulation experiments, it is known that intelligent routing occupies fewer computing resources and has higher throughput than ordinary routing solutions. The algorithm comprehensively considers the impact on the network under high load environment, can timely select the appropriate transmission path, and can realize the collaborative mode of centralized training once and distributed execution multiple times. Compared with other machine learning models, it can quickly find the optimal solution and has the advantages of fast response speed. However, the intelligent routing system only selects the best path for routing decision each time. When the access scale is small, the default routing solution will be applied. In future work, we will continue to study how to reduce the complexity of the model and reduce the training time and computing resources in complex situations and will further study deep reinforcement learning (such as reinforcement learning deterministic policy gradient DPG, proximal policy optimization PPO, etc.) and multi-agent reinforcement learning routing solutions in cloud computing data center networks.

## Author Contributions

**Data curation:** Song Qian.

**Formal analysis:** Tianping Zhang.

**Project administration:** Song Qian.

**Resources:** Song Qian.

**Writing – original draft:** Tianping Zhang, Siping Hu.

**Writing – review & editing:** Tianping Zhang, Siping Hu.

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
