## [Decision Letter · Decision Letter 0]

12 Nov 2024

PONE-D-24-42537Research on Intelligent Routing in the Internet of BodyPLOS ONE

Dear Dr. Zhang,

Thank you for submitting your manuscript to PLOS ONE. After careful consideration, we feel that it has merit but does not fully meet PLOS ONE’s publication criteria as it currently stands. Therefore, we invite you to submit a revised version of the manuscript that addresses the points raised during the review process.

We look forward to receiving your revised manuscript.

Kind regards,

Zeheng Wang

Academic Editor

PLOS ONE

Journal Requirements: When submitting your revision, we need you to address these additional requirements. 1. Please ensure that your manuscript meets PLOS ONE's style requirements, including those for file naming. The PLOS ONE style templates can be found at https://journals.plos.org/plosone/s/file?id=wjVg/PLOSOne_formatting_sample_main_body.pdf and https://journals.plos.org/plosone/s/file?id=ba62/PLOSOne_formatting_sample_title_authors_affiliations.pdf 2. Please note that PLOS ONE has specific guidelines on code sharing for submissions in which author-generated code underpins the findings in the manuscript. In these cases, all author-generated code must be made available without restrictions upon publication of the work. Please review our guidelines at https://journals.plos.org/plosone/s/materials-and-software-sharing#loc-sharing-code and ensure that your code is shared in a way that follows best practice and facilitates reproducibility and reuse. 3. In the online submission form, you indicated that "Researchers can send me E-mail to get some data in the draft." All PLOS journals now require all data underlying the findings described in their manuscript to be freely available to other researchers, either 1. In a public repository, 2. Within the manuscript itself, or 3. Uploaded as supplementary information.This policy applies to all data except where public deposition would breach compliance with the protocol approved by your research ethics board. If your data cannot be made publicly available for ethical or legal reasons (e.g., public availability would compromise patient privacy), please explain your reasons on resubmission and your exemption request will be escalated for approval. 4. PLOS requires an ORCID iD for the corresponding author in Editorial Manager on papers submitted after December 6th, 2016. Please ensure that you have an ORCID iD and that it is validated in Editorial Manager. To do this, go to ‘Update my Information’ (in the upper left-hand corner of the main menu), and click on the Fetch/Validate link next to the ORCID field. This will take you to the ORCID site and allow you to create a new iD or authenticate a pre-existing iD in Editorial Manager. 5. Please update your submission to use the PLOS LaTeX template. The template and more information on our requirements for LaTeX submissions can be found at http://journals.plos.org/plosone/s/latex. 6. Please ensure that you refer to Figure 3 in your text as, if accepted, production will need this reference to link the reader to the figure.

**Additional Editor Comments:**

Please refer to the enclosed comments from the reviewers in your revision. Pay attention to the reference list to ensure that the forms obey the journal's requirements.

Reviewers' comments:

Reviewer's Responses to Questions

**Comments to the Author**

1. Is the manuscript technically sound, and do the data support the conclusions?

Reviewer #1: Yes

Reviewer #2: Yes

2. Has the statistical analysis been performed appropriately and rigorously? 

Reviewer #1: Yes

Reviewer #2: Yes

3. Have the authors made all data underlying the findings in their manuscript fully available?

Reviewer #1: Yes

Reviewer #2: Yes

4. Is the manuscript presented in an intelligible fashion and written in standard English?

Reviewer #1: Yes

Reviewer #2: Yes

5. Review Comments to the Author

Reviewer #1: In this paper the authors proposed an artificial intelligence routing algorithm, combines the variational autoencoder (VAE) and the generative adversarial network model (GAN) to construct a VAE-GAN model to generate multiple sets of data to achieve data enhancement in the Internet of Body. The optimization goals are to maximize the throughput of the Internet of Body and minimize the transmission cost. Here are my comments:

1. Title of the paper doesn’t reflect the body or contribution of the work. Title should be revised.

2. Paper has some merit but it is not well written. Very poor type setting.

3. Introduction and related work is incomplete and no flow in the paper. It’s really difficult to read the paper.

4. Even through many recent works in WBAN domain the authors provide only 13 references and most of them are old.

5. Various modeling like channel model, network model is missing in the paper.

6. Algo. 1 is confusing and it did not reflect well in the body of the paper.

7. Simulation parameters are not provided in the paper. Only 3 outputs can not justify the novelty of the work.

8. I suggested to the author to read recent articles and add them in related works. Some suggestion: 10.1109/JSEN.2024.3440412, https://doi.org/10.1016/j.iot.2024.101151, 10.1109/ACCESS.2023.3236403, 10.1109/TCE.2024.3412942, 10.1109/JIOT.2024.3458976, 10.1109/ACCESS.2024.3476424

Reviewer #2: I think the article is excellent, I congratulate the authors. Some suggestions could be considered:

1. Algorithm specificity: Explaining more clearly why VAE and GAN were combined instead of other approaches could add value.

2. Results and conclusions: While it is mentioned that the experiments demonstrated good results, it would be useful to briefly indicate what type of improvement or specific metric was achieved (e.g., latency reduction, increased network efficiency, lower error rate, etc.).

3. Although I do not see it as 100% necessary, I think it would be great to incorporate a comparison with fuzzy logic. This could significantly strengthen the work; I recommend it for the following reasons:

1. Combination with AI: Fuzzy logic is not incompatible with artificial intelligence techniques such as neural networks, GANs, or VAE. In fact, one can explore how systems based on fuzzy logic can complement or be combined with generative models, to handle additional uncertainties in the data generation or route prediction process. 2. Increased Interpretability: One of the problems with AI-based models, such as GANs, is the “black box” they are based on, which makes it difficult to interpret their decisions. Fuzzy logic could offer a way to add interpretability to the model, as routing decisions can be explained in terms of fuzzy rules, such as “If congestion is high and latency is low, then this route is suitable.” This could make your model more understandable and reliable in medical or critical settings.

3. Efficiency and Computational Resources: Fuzzy logic can be computationally lighter than complex models based on deep or generative neural networks, which could be an important advantage if real-time performance and resource efficiency are crucial in the use case (as mentioned in your abstract about the need to use few computational resources).

4. Routing Optimization: Fuzzy systems can be effectively integrated into routing algorithms to optimize network resource usage by dynamically adjusting routing decisions based on context and changing network conditions. You could compare whether fuzzy logic offers an advantage over your VAE-GAN model, especially when dealing with non-deterministic routes and decisions based on multiple imprecise factors.

5. Simplicity and Flexibility: Fuzzy logic allows for simpler and more flexible representation of relationships between variables, without the need for strict, discrete rules. In the case of your work, you could integrate fuzzy logic to handle different degrees of “quality” or “reliability” of routing paths, rather than simply classifying paths as optimal or non-optimal, which could be a more robust approach to varying network conditions.

In summary, how could you integrate this recommendation in your paper: You could include a section in your paper where you briefly describe the advantages of fuzzy logic compared to deep learning-based methodologies (such as VAE-GAN), and discuss how they could complement or compare each other to solve the routing problem in "Internet of the Body" networks.

Additionally, I dare to send you a small idea of a proposal.

"Although the proposed model based on VAE-GAN has shown a significant improvement in real-time route optimization, it is interesting to consider the integration of fuzzy logic as a possible alternative or complement. Fuzzy logic could offer advantages in making more interpretable and robust decisions in the face of uncertain and fluctuating network conditions, such as congestion or variability of sensor data. Furthermore, the simplicity and flexibility of fuzzy logic could reduce computational complexity in situations where resources are limited. Comparing these approaches and exploring their combination could result in even more efficient optimization of routing in the Internet of the Body, especially in dynamic and high-performance scenarios."

6. PLOS authors have the option to publish the peer review history of their article (what does this mean?). If published, this will include your full peer review and any attached files.

Reviewer #1: No

Reviewer #2: **Yes: **Magister Manuel María Batista Rodríguez

---

## [Author Response · Author response to Decision Letter 0]

11 Dec 2024

Response to Reviewers

Dear Reviewers,

Thank you very much for your review. We have carefully revised and improved according to your suggestions. Please refer to the details below. We hope to meet the requirements of the journal. If you have any other questions, please contact us.

Wishing you a happy life and good health.

The authors

These are all Response. （All responses from the authors are presented with a gray background.）

Reviewer #1: In this paper the authors proposed an artificial intelligence routing algorithm, combines the variational autoencoder (VAE) and the generative adversarial network model (GAN) to construct a VAE-GAN model to generate multiple sets of data to achieve data enhancement in the Internet of Body. The optimization goals are to maximize the throughput of the Internet of Body and minimize the transmission cost. Here are my comments:

1. Title of the paper doesn’t reflect the body or contribution of the work. Title should be revised.

1.Author's response: Thank you for the review. The title has been updated. The new title is Research on Intelligent Routing with VAE-GAN in the Internet of Body.

2. Paper has some merit but it is not well written. Very poor type setting.

2.Author's response: Thank you for the review. We have made detailed modifications, please see Revised Manuscript with Track Changes.

3. Introduction and related work is incomplete and no flow in the paper. It’s really difficult to read the paper.

3.Author's response: Thank you for the review. We have made new modifications. And added some paper to RELATED WORK. And Revised the grammar of the entire text.

4. Even through many recent works in WBAN domain the authors provide only 13 references and most of them are old.

4.Author's response: Thank you for the review. We modified and added more paper in REFERENCES.

5. Various modeling like channel model, network model is missing in the paper.

5.Author's response: Thank you for the review. The network model is in manuscript lines 202 – 239.

6. Algo. 1 is confusing and it did not reflect well in the body of the paper.

6.Author's response: Thank you for the review. Figure 1 is to illustrate the application scenario of this manuscript, which is to solve the routing problem of Internet of Body during large-scale access, and to address the usage scenario of h Internet of Body.

7. Simulation parameters are not provided in the paper. Only 3 outputs can not justify the novelty of the work.

7.Author's response: Thank you for the review. We added more simulation parameters and experiments in this manuscript.

8. I suggested to the author to read recent articles and add them in related works. Some suggestion: 10.1109/JSEN.2024.3440412, https://doi.org/10.1016/j.iot.2024.101151, 10.1109/ACCESS.2023.3236403, 10.1109/TCE.2024.3412942, 10.1109/JIOT.2024.3458976, 10.1109/ACCESS.2024.3476424

8.Author's response: Thank you for the review. We modified and added more paper (including the above papers) in REFERENCES.

Reviewer #2: I think the article is excellent, I congratulate the authors. Some suggestions could be considered:

1. Algorithm specificity: Explaining more clearly why VAE and GAN were combined instead of other approaches could add value.

1.Author's response: Thank you for the review. VAE and GAN mainly solve the problem of limited high-value samples, which has been added to the manuscript.

2. Results and conclusions: While it is mentioned that the experiments demonstrated good results, it would be useful to briefly indicate what type of improvement or specific metric was achieved (e.g., latency reduction, increased network efficiency, lower error rate, etc.).

2.Author's response: Thank you for the review. Parameter analysis has been added in the experiment. Like: The normalized throughput tends to stabilize, indicating that the model can apply the influence of high dynamics.

3. Although I do not see it as 100% necessary, I think it would be great to incorporate a comparison with fuzzy logic. This could significantly strengthen the work; I recommend it for the following reasons:

1. Combination with AI: Fuzzy logic is not incompatible with artificial intelligence techniques such as neural networks, GANs, or VAE. In fact, one can explore how systems based on fuzzy logic can complement or be combined with generative models, to handle additional uncertainties in the data generation or route prediction process. 2. Increased Interpretability: One of the problems with AI-based models, such as GANs, is the “black box” they are based on, which makes it difficult to interpret their decisions. Fuzzy logic could offer a way to add interpretability to the model, as routing decisions can be explained in terms of fuzzy rules, such as “If congestion is high and latency is low, then this route is suitable.” This could make your model more understandable and reliable in medical or critical settings.

3. Efficiency and Computational Resources: Fuzzy logic can be computationally lighter than complex models based on deep or generative neural networks, which could be an important advantage if real-time performance and resource efficiency are crucial in the use case (as mentioned in your abstract about the need to use few computational resources).

4. Routing Optimization: Fuzzy systems can be effectively integrated into routing algorithms to optimize network resource usage by dynamically adjusting routing decisions based on context and changing network conditions. You could compare whether fuzzy logic offers an advantage over your VAE-GAN model, especially when dealing with non-deterministic routes and decisions based on multiple imprecise factors.

5. Simplicity and Flexibility: Fuzzy logic allows for simpler and more flexible representation of relationships between variables, without the need for strict, discrete rules. In the case of your work, you could integrate fuzzy logic to handle different degrees of “quality” or “reliability” of routing paths, rather than simply classifying paths as optimal or non-optimal, which could be a more robust approach to varying network conditions.

In summary, how could you integrate this recommendation in your paper: You could include a section in your paper where you briefly describe the advantages of fuzzy logic compared to deep learning-based methodologies (such as VAE-GAN), and discuss how they could complement or compare each other to solve the routing problem in "Internet of the Body" networks.

Additionally, I dare to send you a small idea of a proposal.

"Although the proposed model based on VAE-GAN has shown a significant improvement in real-time route optimization, it is interesting to consider the integration of fuzzy logic as a possible alternative or complement. Fuzzy logic could offer advantages in making more interpretable and robust decisions in the face of uncertain and fluctuating network conditions, such as congestion or variability of sensor data. Furthermore, the simplicity and flexibility of fuzzy logic could reduce computational complexity in situations where resources are limited. Comparing these approaches and exploring their combination could result in even more efficient optimization of routing in the Internet of the Body, especially in dynamic and high-performance scenarios."

---

## [Decision Letter · Decision Letter 1]

29 Dec 2024

PONE-D-24-42537R1Research on Intelligent Routing with VAE-GAN in the Internet of BodyPLOS ONE

Dear Dr. Zhang,

Thank you for submitting your manuscript to PLOS ONE. After careful consideration, we feel that it has merit but does not fully meet PLOS ONE’s publication criteria as it currently stands. Therefore, we invite you to submit a revised version of the manuscript that addresses the points raised during the review process.

We look forward to receiving your revised manuscript.

Kind regards,

Zeheng Wang

Academic Editor

PLOS ONE

Journal Requirements:

Reviewers' comments:

Reviewer's Responses to Questions

**Comments to the Author**

1. If the authors have adequately addressed your comments raised in a previous round of review and you feel that this manuscript is now acceptable for publication, you may indicate that here to bypass the “Comments to the Author” section, enter your conflict of interest statement in the “Confidential to Editor” section, and submit your "Accept" recommendation.

Reviewer #1: All comments have been addressed

Reviewer #3: All comments have been addressed

2. Is the manuscript technically sound, and do the data support the conclusions?

Reviewer #1: Partly

Reviewer #3: Partly

3. Has the statistical analysis been performed appropriately and rigorously? 

Reviewer #1: Yes

Reviewer #3: Yes

4. Have the authors made all data underlying the findings in their manuscript fully available?

Reviewer #1: Yes

Reviewer #3: Yes

5. Is the manuscript presented in an intelligible fashion and written in standard English?

Reviewer #1: Yes

Reviewer #3: Yes

6. Review Comments to the Author

Reviewer #1: Thanks for the revision. Authors address all of my comments. Current version of the paper is better than previous version.

Reviewer #3: 1. The necessity to design intelligent routing algorithms in the internet of body should be well clarified, since some other kinds of methods can be also used for routing designs.

2. The recent Q-learning based routing scheme in UAV networks can be also considered for comparison with this work#Deleted according to Editorial Policy#.

3. In the related works, the authors are suggested to explain discuss the Transfer learning.

4. The scenario and system model for IOB are not clear in the third section.

5. The algorithm analysis should be well provided.

7. PLOS authors have the option to publish the peer review history of their article (what does this mean?). If published, this will include your full peer review and any attached files.

Reviewer #1: No

Reviewer #3: No

---

## [Author Response · Author response to Decision Letter 1]

1 Jan 2025

Response to Reviewers

Dear Reviewers,

Thank you very much for your review. We have carefully revised and improved according to your suggestions. Please refer to the details below. We hope to meet the requirements of the journal. If you have any other questions, please contact us.

Wishing you a happy life and good health.

The authors

2nd January, 2025

These are all Response. （All responses from the authors are presented with a gray background.）

Reviewer #1: Thanks for the revision. Authors address all of my comments. Current version of the paper is better than previous version.

Reviewer #3: 1. The necessity to design intelligent routing algorithms in the internet of body should be well clarified, since some other kinds of methods can be also used for routing designs.

Author's response:

Thank you very much for your valuable feedback. We have added multiple reasons for using intelligent routing and compared it with other directions. 

We will add its necessity in the first section.

Low intelligence: Existing routing algorithms have the disadvantage of low intelligence in the Internet of Things. With the emergence of technologies such as cloud computing and high concurrency, there is a need for more intelligent routing algorithms to adapt to them.

2. The recent Q-learning based routing scheme in UAV networks can be also considered for comparison with this work#Deleted according to Editorial Policy#.

Author's response:

Thank you very much for your valuable feedback.

In section four, we compared reinforcement learning RL with literature [23], and the Q-learning you mentioned is similar to this work. Therefore, in the normalized transmission delay section of Section 4, we compared the Q-learning you mentioned.

3. In the related works, the authors are suggested to explain discuss the Transfer learning.

Author's response:

Thank you very much for your valuable feedback.

We have completed the explanation of adding transfer learning in the related work.

Transfer learning is a machine learning method that takes the model developed for task1 as an initial point and reuses it in the process of developing the model for task2. It means transferring knowledge from one domain (source domain) to another domain (target domain) to accelerate the learning process of the new domain. This ability to draw analogies is not only in line with human learning laws, but also one of the important goals pursued by artificial intelligence. In deep learning, transfer learning achieves fast adaptation and efficient learning by utilizing pre trained models and combining them with a small amount of new domain data.

4. The scenario and system model for IOB are not clear in the third section.

Author's response:

Thank you very much for your valuable feedback.

In the design of the third section, we studied papers [13-15] and simulated the entire IOB as a network like G=(V,N). And design routing algorithms based on the data transmission methods of the IOB.

5. The algorithm analysis should be well provided.

Author's response:

Thank you very much for your valuable feedback.

The designed algorithm focuses on low computation and space complexity and is suitable for application in the Internet of body. The entire algorithm consists of 2 loops, with the number of loops being tn1 (lines 02--09) and tn2 (lines 11--16) respectively. The time complexity of the entire algorithm is O(tn1)+O(tn2). If the number of loops is n, then the time complexity of the algorithm is O(n).

---

## [Editor Report · Decision Letter 2]

3 Jan 2025

Research on Intelligent Routing with VAE-GAN in the Internet of Body

PONE-D-24-42537R2

Dear Dr. Zhang,

We’re pleased to inform you that your manuscript has been judged scientifically suitable for publication and will be formally accepted for publication once it meets all outstanding technical requirements.

Kind regards,

Zeheng Wang

Academic Editor

PLOS ONE
---

## [Editor Report · Acceptance letter]

8 Jan 2025

PONE-D-24-42537R2 

PLOS ONE

Dear Dr. Zhang, 

I'm pleased to inform you that your manuscript has been deemed suitable for publication in PLOS ONE. Congratulations! Your manuscript is now being handed over to our production team.

Kind regards, 

on behalf of

Dr. Zeheng Wang 

Academic Editor

PLOS ONE